# Expression and Prognostic Value of a Novel B7-H3 (CD276) Antibody in Acute Myeloid Leukemia

**DOI:** 10.3390/cancers16132455

**Published:** 2024-07-04

**Authors:** Sylwia A. Stefańczyk, Clara Hayn, Jonas Heitmann, Susanne Jung, Latifa Zekri, Melanie Märklin

**Affiliations:** 1Clinical Collaboration Unit Translational Immunology, German Cancer Consortium (DKTK), Department of Internal Medicine, University Hospital Tuebingen, 72076 Tuebingen, Germany; sylwia-anna.stefanczyk@med.uni-tuebingen.de (S.A.S.); haynclara@gmail.com (C.H.); jonas.heitmann@med.uni-tuebingen.de (J.H.); susanne.jung@med.uni-tuebingen.de (S.J.);; 2Cluster of Excellence iFIT (EXC 2180) ‘Image-Guided and Functionally Instructed Tumor Therapies’, Eberhard Karls University Tuebingen, 72076 Tuebingen, Germany; 3Department of Peptide−based Immunotherapy, Institute of Immunology, University and University Hospital Tuebingen, 72076 Tuebingen, Germany

**Keywords:** B7-H3, acute myeloid leukemia, monoclonal antibody, prognostic biomarker

## Abstract

**Simple Summary:**

We evaluated a novel proprietary murine B7-H3 antibody 8H8 for flow cytometric analyses of patients with acute myeloid leukemia (AML). The results showed substantial B7-H3 expression in a significant proportion of AML cases, particularly in the M5 group and in those with intermediate or poor risk profiles. The correlation between high B7-H3 expression and poorer overall and progression-free survival suggests that B7-H3 may serve as an additional promising negative prognostic marker to be considered in treatment decisions in AML.

**Abstract:**

Despite recent advances in immunophenotyping, the prognosis of acute myeloid leukemia (AML) is still mainly estimated using age and genetic markers. As the genetic heterogeneity of AML patients is high, flow cytometry-based classification with appropriate biomarkers can efficiently complement risk stratification and treatment selection. An increased expression of B7-H3 (CD276), an immune checkpoint protein, has been reported and associated with poor prognosis. However, the available data are limited and heterogeneous. Here, we used a novel, proprietary murine anti-B7-H3 8H8 antibody for the flow cytometric analysis of B7-H3 expression in AML blasts from 77 patients. Our antibody reliably detected substantial B7-H3 expression in 62.3% of AML patients. B7-H3 expression was higher in the monocytic French–American–British (FAB) M5 group and in intermediate and poor risk patients according to the European Leukemia Network. Using receiver operating characteristics (ROCs), we identified a specific fluorescence intensity cut-off of 4.45 to discriminate between B7-H3^high^ and B7-H3^low^ expression. High B7-H3 expression was associated with shorter overall survival (OS) and progression-free survival (PFS). In conclusion, we have developed a novel B7-H3 antibody that serves as a new tool for the detection of B7-H3 expression in AML and may help to facilitate risk stratification and treatment selection in AML patients.

## 1. Introduction

Acute myeloid leukemia (AML) is a highly malignant disease of the bone marrow, characterized by rapid proliferation of aberrant, immature myeloid cells [1]. It is the most common form of acute leukemia in adults, with a median age at diagnosis of 65–70 years [1,2,3,4]. The prognosis is variable and depends on patient-related factors (age and white blood cell (WBC) count at diagnosis) and AML-related factors (gene mutations such as FLT3-ITD, NPM1 and CEBPA, among others, and cytogenetic aberrations such as t(8;21), inv(16) and t(15;17)) [3,5,6]. According to the 2022 European Leukemia Network (ELN), patients are classified into three risk categories (favorable, intermediate, poor) based on their genetic profile [6,7]. Furthermore, flow cytometry-based immunophenotyping has been introduced as a complementary tool for AML classification and risk stratification [5]. The identification of novel biomarkers for such immunophenotyping may facilitate treatment selection and better prognosis estimation of AML patients. B7-H3 (CD276) is a type 1 transmembrane protein and, together with CD80, PD-L1 and CTLA-4, is a member of the B7 superfamily of immunoregulatory proteins. It is expressed by antigen-presenting immune cells and in many forms of solid and hematologic malignancies. Its expression is higher in malignant tissues than in healthy tissues and it is therefore considered a tumor-associated antigen. Furthermore, it promotes tumorigenesis and progression and is discussed as a negative prognostic marker in cancer patients [8,9,10,11,12,13,14,15,16,17,18]. Its increased expression in cancer tissues and its function as a co-inhibitory checkpoint protein make B7-H3 an attractive diagnostic marker and target for novel anti-cancer immunotherapies [8]. However, its expression pattern and clinical significance, in terms of prognosis and as a novel therapeutic target, have mainly been investigated in solid tumors and only rarely in hematological malignancies. Furthermore, conclusions regarding its prognostic value in AML patients are inconsistent.

Previously, we developed and characterized a novel B7-H3 antibody targeting a membrane distal epitope (8H8) of the B7-H3 protein [19] (patent application EP3822288A1). The therapeutic efficacy of our novel 8H8 clone has been confirmed in several malignancies, including sarcomas [20] and gastrointestinal cancers [19,21]. Recently, we also demonstrated its therapeutic potential in AML [22]. Here, we used our new 8H8 clone for the clinical evaluation of B7-H3 expression in AML blasts from 77 patients in relation to prognosis and clinical outcome. Expression levels were determined by flow cytometric analysis and correlated with disease outcome and other established genetic markers. We propose the detection of B7-H3 with our new clone 8H8 as a promising novel biomarker for efficient immunophenotyping and suggest the use of the specific fluorescence intensity (SFI) cut-off of 4.45 to discriminate between B7-H3^high^ and B7-H3^low^ expression in order to improve risk stratification of AML patients.

## 2. Materials and Methods

### 2.1. Patient Samples

Seventy-seven peripheral blood samples were collected from patients with AML at primary diagnosis. Peripheral blood mononuclear cells (PBMCs) were isolated by density gradient centrifugation and analyzed by flow cytometry. The median follow-up for all patients was 22.2 months (IQR 1.11–26.7 months). Diagnosis and cytologic classification of the AML samples were based on the morphology and cytochemistry of the bone marrow according to the French–American–British (FAB) classification [23,24]. Risk stratification was performed based on (a) age and (b) WBC count at diagnosis, (c) according to the European Leukemia Network scheme [6,7] and according to (cyto)genetic aberrations. Cytogenetic and molecular analyses were performed at the University of Ulm using standard methods.

### 2.2. Antibody Production

The murine monoclonal anti-B7-H3 antibody (clone 8H8) was generated as previously described [19,25]. Briefly, female BALB/c mice (Charles River, Wilmington, MA, USA) were immunized with a recombinant 4IgB7-H3-Fc fusion protein containing Leu29– Thr461 (Q5ZPR3-2). The variable domains of 8H8 were codon optimized for transfection into CHO cells and inserted into a murine Igγ1k backbone-based IgGsc molecule with C-terminal scFvs [25,26]. The ExpiCHO cell system (Gibco, Carlsbad, CA, USA) was used for antibody production, according to the manufacturer’s recommendations. Subsequently, antibodies were purified from culture supernatants by affinity chromatography on MabSelect^TM^ affinity columns. To ensure quality and purity, the antibodies produced were further subjected to analytical and preparative size exclusion chromatography using Superdex S200 Increase 10/300 GL and HiLoad 16/60 columns (GE Healthcare (Chicago, IL, USA)) and only the fractions containing the monomeric form were used.

### 2.3. Flow Cytometry

PBMCs from AML patients (0.5 × 10^6^ cells per staining) were incubated in medium containing human IgG (10 μg/mL, Sigma-Aldrich, St. Louis, MO, USA) to prevent unspecific Fc receptor binding before staining, then washed and incubated with unconjugated B7-H3 mAb (mouse clone 8H8) or the isotype control at 10 μg/mL, followed by goat antimouse PE-conjugated antibody (1:200, Jackson ImmunoResearch, West Grove, PA, USA). After staining for B7-H3 expression, AML blasts in the PBMCs were identified according to the immune phenotype obtained at diagnosis by simultaneous staining for CD33-BV510, CD34-BV421, CD38-FITC and CD117-PE-Cy7, with an antibody staining cocktail. Fluorescence Ab conjugates (CD33, CD34, CD38, CD117; BioLegend, San Diego, CA, USA) were used at a dilution of 1:100–1:300. Dead cells were excluded based on 7-AAD (1:200, BioLegend, San Diego, CA, USA) positivity. Specific fluorescence intensity was calculated by dividing the median fluorescence obtained with the anti-B7-H3 mAb by the median fluorescence obtained with the IgG1 isotype control. Positive expression was defined as an SFI ≥ 1.5. Measurements were performed using an LSR Fortessa or a FACSCanto II (BD Biosciences, Heidelberg, Germany). Data analysis was performed using FlowJo_V10 software (FlowJo LCC, Ashland, OR, USA).

### 2.4. Statistical Analysis

Data are presented as mean ± SD. Scatter dot plots include the median and the interquartile range (IQR). The unpaired Student’s *t*-est, Mann–Whitney-/Kruskal–Wallis test, Chi^2^ test or Fisher’s exact test was used to compare groups. The appropriate test was selected according to the data distribution and group size. The distribution of overall survival (OS) was calculated using the Kaplan–Meier method. The log-rank test was used to compare survival between groups. The correlation between B7-H3 SFI and B7-H3-positive cells (%) was analyzed using Spearman’s rank order correlation. For multivariate analysis, Cox’s proportional hazards regression was performed. Patients with missing data were excluded from the respective analyses.

To determine the predictive cut-off value, we sub-grouped the B7-H3 SFI with respect to corresponding OS times and by the treatment used. Receiver operating characteristic (ROC) analysis was performed using JMP^®^ Pro. The value of the highest Youden index (YI) was determined. Comparison of the obtained YIs with the overall B7-H3 SFI distribution revealed the identity of the YIs at the lower boundary of the 4th quartile. Therefore, for the sake of comparability, the lower boundary of the 4th quartile (4.45) was selected as the B7-H3 SFI cut-off value and used to separate patient samples into B7-H3^high^ and B7-H3^low^ in all further analyses.

All statistical analyses were performed with JMP^®^ Pro (version 16.2.0 for Windows, ©SAS Institute Inc. (Cary, NC, USA)). *p* values of < 0.05 were considered statistically significant. All figures were generated using GraphPad Prism (version 9.4.1 for Windows, ©GraphPad Software Inc. (La Jolla, CA, USA)).

## 3. Results

### 3.1. Clinical Characterization of the AML Patients

B7-H3 expression was determined by flow cytometric analysis of AML samples from 77 patients. All blood samples were collected and analyzed within the first two weeks after diagnosis. The clinical characteristics of the patient cohort are shown in Table 1 and Appendix A. The patient age at diagnosis ranged from 26 to 89 years (mean 70.0 years) with a female/male ratio of 30:47. Three-quarters of the patients were diagnosed with primary AML and one-quarter were diagnosed with secondary AML. A total of 23 (23) patients presented with undifferentiated leukemia (30% of all, n_M0_ = 7, n_M1_ = 16), 17 with granulocytic leukemia (23% of all, n_M2_ = 11, n_M3_ = 6) and 35 with monocytic leukemia (47% of all, n_M4_ = 17, n_M5_ = 18). Two patients could not be classified according to the French–American–British (FAB) classification. According to the 2016 WHO classification [27], 10 patients had AML with myelodysplasia-related changes (13.0%), 44 had AML with recurrent genetic aberrations (57.0%), 21 had AML that was not otherwise specified (27.0%) and 2 had therapy-related myeloid neoplasms (3.0%). Regarding prognosis, 25 patients were classified as favorable (38.5%), 28 as intermediate (43.0%) and 12 as poor (18.5%) according to the 2022 ELN risk stratification.

The cytogenetic analysis identified 7 cases with complex karyotype aberrations, 23 cases with <3 aberrations, 6 cases with t(15;17), 1 case with t(8;21) and 3 cases with inv(16). FLT3-ITD, NPM1 and CEBPA mutations were detected in 28, 28 and 5 patients, respectively (Appendix A).

At initial diagnosis, the mean white blood cell (WBC) count was 110.9 G/L with a range of 4.6–448.3 G/L. The mean hemoglobin (Hb) at initial diagnosis was 8.6 g/dL (range 3.8–12.9 g/dL) and platelet count (Plt) was 78.0 G/L (range 6.0–433.0 G/L). The mean percentage of leukemic blasts in the peripheral blood (PB) of the included cases (*n* = 63) was 80.4% (range 23–100%), and in the patients’ bone marrow (BM, *n* = 36), it was 74.7% (range 12–98%).

Of the 77 patients, 55 (71.0%) received at least one induction therapy and 22 (29.0%) received non-intensive treatment, which included the best supportive care (symptomatic treatment), hydroxyurea, low-dose cytarabine (Ara-C) and hypomethylating agents. Of the patients who received induction therapy, 46 (83.6%) received anthracycline, while 6 (10.9%) received a non-anthracycline-based regimen. For three patients (5.5%), the first induction therapy regimen is unknown. Of the patients who received induction, 50 went on to consolidation. In this group, 29 patients received a consolidation regimen, while 18 patients received a cytarabine-only regimen. The response to the first induction was defined according to the European Leukemia Network (ELN) definition [6,7]. Response to the first induction with complete remission with or without incomplete hematologic recovery (CR(i)) was observed in 71.1% of patients (*n* = 31). A partial remission (PR) was observed in 28.9% of patients. Of all patients, 29 (38.0%) received an allogeneic hematopoietic-cell transplantation (allo-HCT).

Before proceeding with our analysis, we ensured the representativeness of our selected AML patient cohort by analyzing the overall survival (OS). Here, we showed that, as expected, the OS was significantly longer in patients who received first induction therapy (median OS 16.95 months) compared to those who received non-intensive treatment only (median OS 0.75 months) (Appendix A).

### 3.2. B7-H3 Expression on AML Blasts

The AML blasts were analyzed for B7-H3 expression by flow cytometry using a mouse anti-human B7-H3 8H8 mAb. The gating strategy for a patient with high and low B7-H3 expression is shown in Figure 1A. The percentage of B7-H3-positive cells varied widely, ranging from 0.1% to 99.9% with a mean of 32.6% of all AML blasts (Figure 1B). Using expression intensity, defined as SFI levels and considered relevant with SFI levels ≥1.5, highly variable surface levels of B7-H3 were observed with up to an SFI of 19.9, with a mean SFI of 4.0 (Figure 1C). In summary, relevant B7-H3 levels were expressed by 62.3% of all patient samples tested (Figure 1D). As expected, B7-H3 SFI levels showed a strong and positive correlation with the percentage of B7-H3-positive cells (Figure 1E; Spearman’s rank correlation coefficient = 0.899, *p* <0.0001 ***, *n* = 77).

### 3.3. B7-H3 Expression and Clinical and Genetic Characteristics of AML Patients

Regarding clinical characteristics, B7-H3 expression levels did not differ significantly between primary and secondary AML samples, nor between samples obtained from patients younger and older than 60 years at diagnosis (Figure 2A). For FAB subtypes, B7-H3 expression showed heterogeneous differences; in particular, samples from the M5 subtypes showed significantly higher B7-H3 SFI values (mean 6.3 ± 5.3) compared to the M4 (mean SFI 3.2 ± 3.3), M2 (mean SFI 1.6 ± 1.1) and M1 (mean SFI 3.7 ± 5.4) subtypes (Figure 2B). FAB subtype clustering revealed a significantly higher B7-H3 SFI in monocytic lineage samples (M4–M5), compared to undifferentiated and granulocytic lineage samples (M0–M2) (Figure 2C; mean SFI_M4–M5_ 4.7 ± 4.4; mean SFI_M0–M2_ 3.2 ± 4.2, *p* = 0.005). The expression intensity of B7-H3 was observed to vary significantly on the surface of AML CD34 cells, but did not show significant differences between CD34− and CD34+ AML samples (Figure 2D).

Regarding genetic background, B7-H3 SFI did not differ significantly between samples from patients with or without karyotypic or cytogenetic aberrations [t(8;21), t(15;17), inv(16)], nor with or without FLT3-ITD or NPM1 mutations. However, the SFI levels differed significantly between samples from CEBPA wild-type (wt) and CEBPA mutant (mt) patients (Figure 2E, mean SFI_wt_ 4.2 ± 4.8, mean SFI_mt_ 1.6 ± 1.5, *p* = 0.034). Unless otherwise noted, the acute promyelocytic leukemia (APL) FAB M3 group was excluded from the following analyses due to the significantly better prognosis and different treatment regimens typically observed in APL patients. This could potentially bias the results. According to the ELN 2022 risk stratification, the B7-H3 SFI differed between samples from patients with a favorable (fav) risk compared to those with intermediate (int) and poor (pr) risks (Figure 2F; SFI_fav_ = 2.4 ± 2.1, SFI_int+pr_ = 4.9 ± 5.1, *p* = 0.029).

### 3.4. Association of B7-H3 Expression with Clinical Outcome

Finally, we analyzed whether and how overall survival (OS) differed between patients according to the B7-H3 SFI quartiles (Figure 2G). We found that OS differed significantly among the quartile groups and that the differences in OS were greatest between the 4th quartile and the remaining pooled quartile groups. Therefore, we performed a group comparison, which showed a statistically significant difference in OS between the 1st and 4th quartile (median OS of 1st quartile: 2.1 months; median OS of 4th quartile: 18.4 months; HR = 0.42 [95% CI 0.18–0.96], *p* = 0.022).

For precision and ease of use, we performed an ROC analysis to determine an SFI cut-off of 4.45 to discriminate between high (B7-H3^high^) and low (B7-H3^low^) expression of B7-H3 on AML blasts. Using this cut-off, we performed a groupwise comparison, which showed that patients with B7-H3^high^ expression at diagnosis did not differ significantly from patients with B7-H3^low^ expression in terms of their age, sex and AML occurrence (primary/secondary) distribution, nor in their WBC, Hb and Plt counts (Table 2). In addition, although not significant, the ELN 2022 risk groups showed a slightly different distribution between B7-H3^high^ and B7-H3^low^ patients, with fewer patients being stratified as having a favorable risk (25.0% vs. 40.8%) and more as having an intermediate risk (75.0% vs. 42.9%) in the B7-H3^high^ patients compared to the B7-H3^low^ patients (Figure 2H, Table 2). Similar results were found for the FAB subgroup distribution; overall, the differences between B7-H3^high^ and B7-H3^low^ patients were not significant. However, the proportions of M2 and M5 subtypes differ in that the proportion of M5 was higher (44.4% vs. 17.5%), whereas the proportion of M2 was lower (5.6% vs. 17.5%) in B7-H3^high^ compared to B7-H3^low^ patient samples (Figure 2I, Table 2). The remaining FAB subtypes showed similar proportions. Regarding the response to the first induction therapy, there were no significant differences between B7-H3^high^ and B7-H3^low^ patients. Descriptively, however, the proportion of CR(i) was higher in B7-H3^high^ (87.5%) compared to B7-H3^low^ (67.6%) patients (Table 2). Finally, regarding cytogenetic and genetic aberrations, there were no significant differences in their distribution between B7-H3^high^ and B7-H3^low^ patients (Appendix A).

OS and PFS differed significantly between patients with B7-H3^high^ and those with B7-H3^low^ expression on AML blasts, regardless of their treatment regimen. Thus, the OS and PFS were longer in the B7-H3^low^ group (median OS 16.8 months) compared to the B7-H3^high^ group (median OS 2.1 months; Figure 3A,B; HR_OS_ = 0.49 [95%CI 0.23–0.95], p_OS_ = 0.021, HR_PFS_ = 0.47 [95%CI 0.21–0.94], p_PFS_ = 0.016). Furthermore, patients who received allogeneic stem cell transplantation showed a longer OS if they were in the B7-H3^low^ group (median OS 32.9 months) compared to those in the B7-H3^high^ group (median OS 12.1 months) (Appendix A; HR = 0.03, [95%CI 0.04–1.06], *p* = 0.043).

As the proportions of the different treatment regimens was not similarly distributed in the B7-H3^high^ and B7-H3^low^ cases (Table 2), two subgroup analyses were performed after grouping the patients according to their therapy. Among the patients who received an anthracycline-based first induction therapy, those with B7-H3^low^ AML blasts showed a significantly longer OS (median OS 138.8 months) than those with B7-H3^high^ AML blasts (median OS 16.8 months; Figure 3C; HR = 0.38 [95%CI 0.08–0.81 p = 0.019). In contrast, there was no significant difference in OS between the B7-H3^low^ and B7-H3^high^ groups in patients receiving non-intensive treatment only (0.6 months vs. 1.0 months; Figure 3D; HR = 0.55 [95%CI 0.21–1.82], *p* = 0.167).

Finally, we performed Cox’s proportional hazards regression analysis on the data from all patients receiving first induction therapy (55, of whom, 6 received a non-anthracycline-based regimen), and identified high B7-H3 expression on the AML blasts together with age ≥60 years at diagnosis and a WBC of ≥100,000 G/L at diagnosis as independent risk factors for poor outcome (i.e., shorter time to death after diagnosis). In contrast, the analysis revealed that the ELN 2022 favorable risk classification and mutated CEBPA reduced the risk for a short OS after diagnosis (Figure 4).

## 4. Discussion

B7-H3 is currently under intense investigation as both a promising prognostic marker and a target for cancer immunotherapy. Most studies have investigated B7-H3 expression and its targeting effects in solid tumor entities, while only a few studies have focused on AML [16,17,18,28,29,30,31,32,33,34,35,36,37]. The elevated expression in malignant tissues, as well as its immunoinhibitory function that promotes tumor immune escape, qualifies B7-H3 as an attractive molecule for therapeutic targeting [8,30].

In this study, we used our novel proprietary 8H8 antibody to characterize B7-H3 expression on AML blasts and to determine whether B7-H3 expression is associated with clinical outcomes in these patients. We detected an overall high level of B7-H3 expression in approximately two-thirds of all patient samples tested, with heterogeneity depending on the ELN risk group, genetic background as well as the lineage of origin of the AML blasts (i.e., FAB subgroup). Our results are broadly consistent with previously published data from Guery et al. and Hu et al., with slight variations discussed below [16,17]. Finally, we confirmed that B7-H3 is an emerging negative prognostic marker in AML and established an SFI cut-off value of 4.5, which, together with established prognostic values, allows its application for AML risk stratification.

Our data showed that B7-H3 expression is high in AML blasts, with 63.3% of all AML patient samples having an SFI ≥1.5 and thus being positive. Previously published data on the proportion of AML patients expressing B7-H3 vary from 27%, over 44.8%, to up to 60%, with a 2- to 3-fold higher expression in AML compared to healthy samples [16,17,31,33]. Thus, our results stand in agreement with these reports. Furthermore, we demonstrated that B7-H3 expression levels are independent of the AML occurrence (primary/secondary), gender, age and WBC at diagnosis, as reported by Guery et al. (2015) [16]. In addition, the B7-H3 expression level was independent of commonly used cytogenetic and monogenetic risk markers, namely karyotype aberrations, t(15:17), inv(16), as well as FLT-ITD and NPM1 mutation status. All of these results, together with the Cox’s proportional hazards regression results, indicate that B7-H3 expression is an independent negative prognostic marker with broad applicability.

Regarding FAB subtypes, our data showed that samples from the FAB M5 patients have the highest B7-H3 SFI. Consistent with our results, Zhang, L. et al. (2021) and Zhang, W. et al. (2021) found the highest B7-H3 expression in the FAB M5 group, while Guery et al. (2015) and Lichtman et al. (2021) found the highest B7-H3 expression in the M3 and M5 groups [16,30,34,36]. This is further confirmed by Appendix A, where among all B7-H3^high^ patients, the proportion of the M5 subgroup was the largest (44.4%) and higher compared to the corresponding proportion of B7-H3^low^ patients (17.5%). Clustering of the FAB subgroups according to their lineage of origin confirmed that the B7-H3 SFI was highest in the M4/M5 group. According to the FAB classification system, M4 and M5 comprise cells of the monocytic lineage [23]. Therefore, our results are consistent with the finding that B7-H3 is more prominently expressed on monocytic AML blasts [16,30,34,36]. The fact that B7-H3 expression is mainly increased in monoblastic/monocytic leukemias could be biologically based on the monocyte population. While monocytes do not constitutively express B7-H3 in vivo, several cytokines such as IFNγ are likely to induce B7-H3 expression on monocytes [12,38].

Among our patients, those expressing B7-H3 above an SFI of 4.45 (B7-H3^high^) showed a reduced overall and progression-free survival compared to those expressing B7-H3 below this cut-off (B7-H3^low^). This was true for all patients before stratification by treatment regimen, as well as when considering only those patients who received anthracycline-based first induction therapy. In contrast, B7-H3 had no prognostic value in patients who received non-intensive treatment only. This can be explained by their significantly reduced OS compared to those who received first induction therapy. The OS times for both groups are of such brevity that the individual time points at which the respective survival probabilities are compared were so close together that the differences did not reach statistical significance. In other words, for patients receiving non-intensive treatment alone, the OS time is of such brevity that the additional difference seen according to B7-H3 expression is negligible.

Finally, since anthracycline-based first induction therapy is the classical and most frequently used treatment regimen for AML patients in our cohort (59.7% of all treatments), but also in general [18], we propose considering B7-H3 as a negative prognostic marker for AML, independent of the treatment regimen. Accordingly, B7-H3 SFI could be routinely determined with our proprietary antibody 8H8 by flow cytometry at the time of initial diagnosis and patient prognosis could be estimated according to the cut-off of 4.45.

One particular correlation deserves further consideration: the distribution of mutant (mt) and wild-type (wt) CEBPA cases differs significantly between B7-H3^high^ and B7-H3^low^ patients, with the B7-H3^high^ group harboring only wt but no mt CEBPA cases. In addition, samples from wild-type CEBPA patients show a significantly higher B7-H3 SFI than samples from patients carrying a CEBPA mutation. The latter is consistent with the findings of Guery et al. (2015) and Zhang et al. (2021) [16,34]. Mutated CEBPA is considered a favorable prognostic marker, as patients carrying ≥1 mutation show an increased OS compared to patients with wt CEBPA [39,40]. This is true for our patient cohort, as the multivariate analysis showed a significantly reduced hazard ratio (HR) for death for mt CEBPA compared to wt CEBPA. Taken together, the CEBPA mutation status and B7-H3 expression levels ideally complement each other for prognostic stratification, as wt CEBPA and high B7-H3 are both individually markers of poor outcome and tend to coexist in the same patient.

The prognostic value of B7-H3 is further supported by its expression among the different ELN risk categories. We showed that samples from patients in the ELN intermediate and poor risk categories have significantly higher B7-H3 expression levels than those in the ELN favorable risk category. This is supported by the ELN risk category distribution among the B7-H3^low^ and B7-H3^high^ patients. Although the distribution was not statistically different, descriptively, the proportion of intermediate and poor risk patients was higher in the B7-H3^high^ group (75.0%) compared to the B7-H3^low^ group (57.2%), while the proportion of favorable risk patients was higher in the B7-H3^low^ group (42.9%) compared to the B7-H3^high^ group (25.0%). This stands in agreement with the findings of Tyagi et al. (2022) and Antohe et al. (2020) [28,33].

## 5. Conclusions

In conclusion, our study extends and consolidates previously published data on B7-H3 expression on AML blasts and its prognostic value therein. We were able to demonstrate that our novel, proprietary 8H8 mAb can be used as a reliable tool for immunophenotyping AML patient samples. In addition, the expression of B7-H3 on AML blasts may help in the assessment of patient prognosis. Finally, our characterization of B7-H3 expression on AML blasts will support the development of novel targeted immunotherapeutics for the treatment of AML.

## Figures and Tables

**Figure 1 cancers-16-02455-f001:**
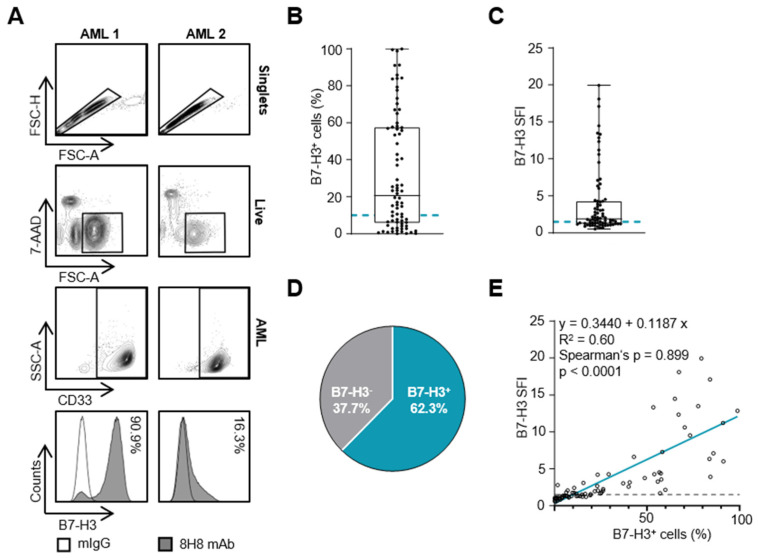
B7-H3 expression on AML blasts. B7-H3 expression on AML blasts was analyzed by flow cytometry. (**A**) Gating strategy for two exemplary AML samples: singlets, viable (7-AAD^−^), blast marker (CD33^+^) and B7-H3 expression as a percentage. The histograms show the B7-H3-specific staining using the murine 8H8 mAb (shaded peaks) and the corresponding isotype control (empty peaks). (**B**,**C**) B7-H3 expression on blasts from AML patients (*n* = 77) is shown as percentage of B7-H3-positive blasts (**B**) and as SFI levels (**C**) (box plots with min/max whiskers). AML SFI levels ≥1.5 and blasts ≥10% B7-H3 expression were considered positive (dotted gray lines). (**D**) Percentage of B7-H3-positive samples according to SFI ≥1.5. (**E**) Spearman correlation of B7-H3 SFI with percentage of B7-H3-positive blasts (Spearman’s rank correlation coefficient (r_s_) = 0.899, *p* < 0.001).

**Figure 2 cancers-16-02455-f002:**
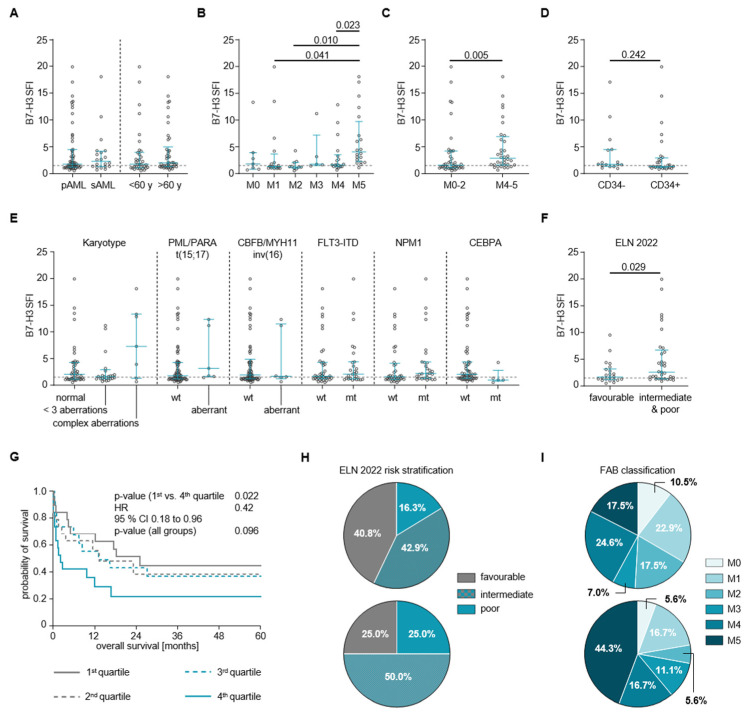
Association of B7-H3 with clinical and genetic parameters. B7-H3 SFI levels according to (**A**) primary (pAML) vs. secondary (sAML) AML and age < 60 and >60 years (single values with median and 25/75 percentile; Mann–Whitney test), (**B**) the different FAB subgroups (single values with median and 25/75 percentile, pairwise Mann–Whitney test), (**C**) FAB M0–2 vs. FAB M4–5 (single values with median and 25/75 percentile; Mann–Whitney test), (**D**) CD34− and CD34+ cell populations, (**E**) cytogenetic [karyotype aberration, t(15;17), inv(16)] and monogenetic [FLT3-ITD, NPM1, CEBPA] markers (single values with median and 25/75-percentile, pairwise Mann–Whitney test) and (**F**) ELN 2022 favorable vs. intermediate/poor risk groups (single values with median and 25/75-percentile; Mann–Whitney test). (**G**) Overall survival (OS) in all patients regardless of their therapy according to B7-H3 SFI quartiles in Kaplan–Meier analysis. Median OS was reached after 2.1 months in the 4th quartile (blue line) and after 18.4 months in the 1st quartile (grey line) (log-rank test, *p* = 0.022). (**H**) Distribution of patients separated by B7-H3^low^ and B7-H3^high^ expression levels among ELN risk groups and among (**I**) FAB subtypes.

**Figure 3 cancers-16-02455-f003:**
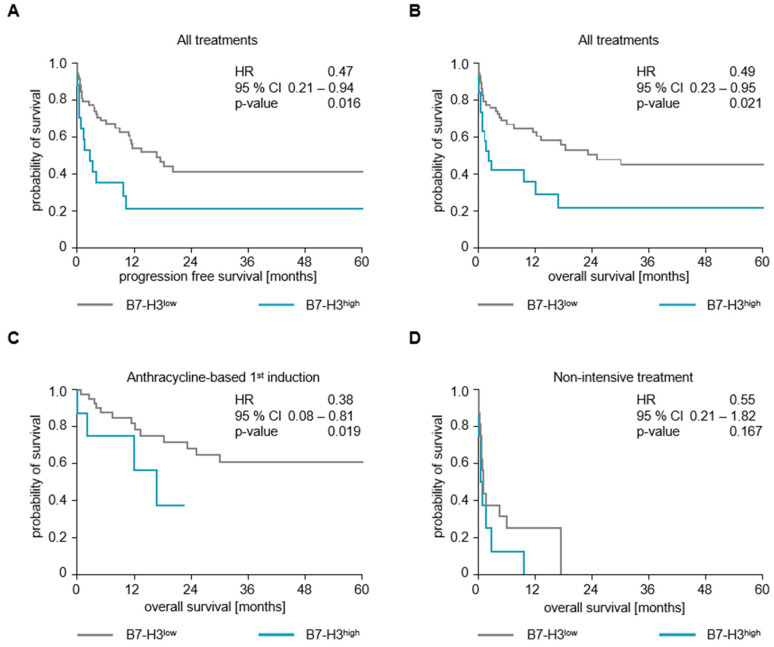
Impact of B7-H3 expression on clinical outcome (**A**) Progression-free survival (PFS) in all patients regardless of their therapy according to B7-H3^low^ and B7-H3^high^ expression in Kaplan–Meier analysis. Median PFS was 2.7 months in B7-H3^high^ (blue line) and 16.8 months in B7-H3^low^ (grey line) (log-rank test, *p* = 0.016). (**B**) Overall survival (OS) in all patients regardless of their therapy according to B7-H3^low^ and B7-H3^high^ expression in Kaplan–Meier analysis. Median OS was reached after 2.1 months in B7-H3^high^ (blue line) and 25.1 months in B7-H3^low^ (grey line) (log-rank test, *p* = 0.021). (**C**) OS in patients with anthracycline-based 1st induction therapy according to B7-H3^low^ and B7-H3^high^ expression in Kaplan–Meier analysis. Median OS was reached after 16.8 months in B7-H3^high^ (blue line) and after 138.8 months in B7-H3^low^ (grey line) (log-rank test, *p* = 0.019). (**D**) OS in patients receiving non-intensive treatment only, according to B7-H3^low^ and B7-H3^high^ expression in Kaplan–Meier analysis. Median OS was 0.6 months in B7-H3^high^ (blue line) and 1.0 months in B7-H3^low^ (grey line) (log-rank test, *p* = 0.167).

**Figure 4 cancers-16-02455-f004:**
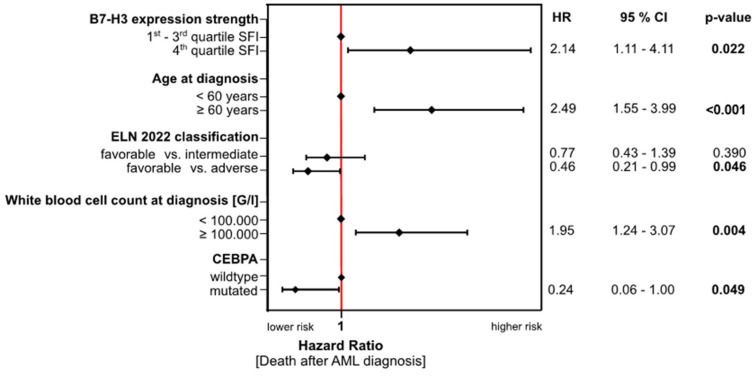
Cox’s proportional hazard regression analysis. Whole model specifications: Chi-Square(5) = 13.18, *p* = 0.0218, *n* = 55. **HR:** hazard ratio; **CI**: confidence interval; **ELN**: European Leukemia Network; **FAB**: French–American–British; **WBC**: white blood cell.

**Table 1 cancers-16-02455-t001:** Patient characteristics.

	Mean	Range
**Age at diagnosis [years]**	60.9	26.0–89.0
**Sex**	***n* = 77**	**% of known**
Female	30	39.0
Male	47	61.0
**Primary/Secondary AML**		
Primary	58	75.3
Secondary	19	24.7
**FAB classification**		
M0	7	9.3
M1	16	21.3
M2	11	14.7
M3	6	8.0
M4	17	22.7
M5	18	24.0
Unknown	2	-
**WHO classification 2016**		
Acute myeloid leukemia with myelodysplasia-related changes	10	13.0
Acute myeloid leukemia with recurrent genetic aberrations	44	57.1
Acute myeloid leukemia, not otherwise specified	21	27.3
Therapy-related myeloid neoplasms	2	2.6
**ELN 2022 classification**		
Favorable	25	38.5
Intermediate	28	43.0
Poor	12	18.5
Unknown	12	-
**Blood count at diagnosis**	**Mean**	**Range**
WBC [G/L]	110.9	4.6–448.3
Hb [g/dL]	8.6	3.8–12.9
Plt [G/L]	78.1	6.0–433.0
**Blasts [%]**	**Mean**	**Range**
PB *n* = 63	80.4	23–100
BM *n* = 36	74.7	12–98

**FAB:** French–American–British; **WHO**: World Health Organization classification according to [27]; **ELN**: European Leukemia Network classification according to [7]; **WBC**: white blood cell; **Hb**: hemoglobin; **Plt**: platelets; **PB**: peripheral blood; **BM**: bone marrow.

**Table 2 cancers-16-02455-t002:** Patients’ characteristics according to B7-H3^low^ and B7-H3^high^ classification.

	B7-H3 Low (SFI < 4.45) (*n* = 58)					B7-H3 High (SFI ≥ 4.45) (*n* = 19)					*p*-Value
	** *n* **	**Mean**	**Range**			** *n* **	**Mean**	**Range**			
Age at diagnosis [years]	58	62.7	26.0	–	89.0	19	67.0	37.0	–	85.0	0.239 ^♦^
WBC [G/L]		65.9	5.0	–	448.3		126.8	4.6	–	361.0	0.116 ^♦^
Hb [g/dL]		9.0	3.8	–	12.9		8.4	4.4	–	10.3	0.190 ^♦^
Plt [G/L]		43.5	6.0	–	433.0		41.0	6.0	–	124.0	0.324 ^♦^
**Sex**		**% within group**					**% within group**				
Female	23	39.7				7	36.8				
Male	35	60.3				12	63.2				0.8274 ^‡^
**Primary/Secondary AML**											
Primary	42	72.4				16	84.2				
Secondary	16	27.6				3	15.8				0.301 ^‡^
**FAB classification**											
M0	6	10.5				1	5.6				
M1	13	22.8				3	16.7				
M2	10	17.5				1	5.6				
M3	4	7.0				2	11.1				
M4	14	24.6				3	16.7				
M5	10	17.5				8	44.4				0.241 ^‡^
Unknown	1	-				1	-				
**WHO classification 2016**											
Acute myeloid leukemia with myelodysplasia-related changes	8	13.8				2	10.5				
Acute myeloid leukemia with recurrent genetic aberrations	31	53.5				13	68.4				
Acute myeloid leukemia, not otherwise specified	17	29.3				4	21.1				
Therapy-related myeloid neoplasms	2	3.5				0	0.0				0.639 ^‡^
**ELN 2022 classification**											
Favorable	21	42.9				4	25.0				
Intermediate	20	40.8				8	50.0				
Poor	8	16.3				4	25.0				0.422 ^‡^
Unknown	9	-				3	-				
**1st Induction**											
No	14	24.1				8	42.1				
Yes	44	75.9				11	57.9				0.132 ^‡^
**Anthracycline**-**based 1st induction**											
No	17	29,8				11	64.7				
Yes	40	70.2				6	35.3				0.007 ^‡^
Unknown	2	-				1	-				
**Response to 1st induction**											
CR(i)	25	67.6				7	87.5				
PR	12	32.4				1	12.5				0.217 ^‡^
Unknown	21	-				11	-				

‡ Pearson’s chi-squared test; ♦ Mann–Whitney U test; **FAB**: French–American–British; **WHO**: World Health Organization classification according to [27]; **ELN**: European Leukemia Network; **WBC**: white blood cells; **Hb**: hemoglobin; **Plt**: platelets; **SFI**: specific fluorescence intensity; **CR(i)**: complete response (with incomplete hematologic recovery); **PR**: partial response, response classification according to ELN 2022 [7].

## Data Availability

The corresponding author had full access to all the data in the study and all authors share final responsibility for the decision to submit for publication. The data sets generated during the current study are available from the corresponding author upon reasonable request.

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
