# Peer review of "Expression and Prognostic Value of a Novel B7-H3 (CD276) Antibody in Acute Myeloid Leukemia"

_cancers, 2024, doi:10.3390/cancers16132455_

Round 1
Reviewer 1 Report
Comments and Suggestions for Authors|
|
|
The article by Stefanczyk et al. deals with the cytofluorimetric evaluation of B7-H3 (CD276) expression by acute myeloid leukemia (AML) cells, and its potential prognostic value.
Overall, it describes a well-conducted retrospective study on this topic, using a novel proprietary antibody developed by the Authors’ Lab.
This is relevant, as B7-H3 belongs to the check-point inhibitors exhibited by AML blasts as a potential tool of immune escape.
I find it, though, that the article could benefit from some changes, especially in the clinical/prognostic part:
1. unfortunately, the Authours decided to include 6 acute promyelocytic leukemia (APL) patients (formerly, AML FAB M3) in their analysis. As APL patients fare significantly better than other types of AML, and benefit in recent years from a substantially different treatment (a combination of All-trans-retinoic acid + Arsenic Trioxide, rather than the combination of Anthracyclin + Citarabine + other chemotherapy), this inclusion may have somehow biased some results, even if the number is small. Later in the article it is detailed that the patients that did not receive “best supportive care” (i.e. transfusions, minor cytoreduction and, basically, palliative care!) received an “anthracycline-based 1st induction” (e.g. see Figure 3), and that should include the 6 APL patients. I would detach these patients from the whole analysis (as it is done in most studies nowadays) or, at least, include more details about the type of therapy these patients received (“anthracycline-based” + ATRA, or more modern ATRA-ATO? Were some of these patients treated only with “best supportive therapy”???).
2. more significantly, 22 patients (29% of the series) did not receive any disease-modifying therapy (i.e. were treated with “best supportive care”, as part of a palliative approach). Unsurprisingly, their survival was dismal, and unaffected by B7-H3 expression (Fig. 3D): other factors, namely disease progression, affected the prognosis more. This is clearly recognized by the Authors in the Discussion (page 12, line 341 and forward), and I agree with their statements. Still, these patients represent a big fraction of the whole (almost one third, in fact), and may have biased the results of the evaluation of the series (although it is clear that when only treated patients were considered, e.g. Fig. 3C, B7-H3 expression still retained prognostic value).
It is not clear to me, though, whether Cox's proportional hazard modeling (multivariate analysis) has been performed on the series as a whole or stratified according to effective treatment. This must be stated clearly in the text, and if Cox’s modeling was done on the whole, this is incorrect and should be repeated using the two groups (or, basically, just the “anthracycline-based 1st induction” group, the one effectively treated with disease modifying approach). Figures and text must be modified accordingly.
3. any prognostic assessment is highly dependent on how the patients have been actually treated. Therefore, the Authours should detail much more the types of treatment that were performed. This is true for APL, but also for all other types of AML. Something is given at page 4, line 158 onwards, but more details are needed on type of induction regimens and consolidation (how many cycles? Based on high-dose/ intermediate dose Cytarabine? Other types of consolidation?). All statistica. analysis should be made separately on patients treated with "best supportive care", intensive chemotherapy, hypomethylating agents (if any) and on APL patients, as well.
Some other comments:
- even if FAB classification is much less relevant today as in the past, it is interesting that B7-H3 seems more expressed by monoblastic/monocytic leukemia. This makes sense, considering the residual minimal differentiation of the blasts and the role of B7-H3 in monocytes/macrophages. Has any specific genetic features / gene mutation been correlated with B7-H3 expression by leukemic blasts? Can the Author speculate why, biologically, expression is different in the Discussion?
- are all patients not on “best supportive care” treated with “anthracycline-based 1st induction”? Have any of the patients in the series been treated with Azacitidine (alone or in combination) or other demethylating agents?
I ask this as there is relevant amount of data pointing out the induction of checkpoint inhibitors by AML blasts by exposure to Azacitidine or other demethylating agents. I understand analysis was made on samples from untreated patients, but was this issue studied during the trial? Is B7-H3 dynamic over disease history, or does it change from diagnosis after chemotherapy (or therapy with demethylating agents)?
Minor changes:
- please note that a couple of times Cox’s proportional hazard modeling is referred as “cox” (no major caps; e.g. page 3, line 120; page 11, line 323); please correct
- some sentences such as “B7-H3 serves as a negative prognosic marker to guide treatment decisions in AML” (end of “simple summary”; same concepts in the “Discussion”) might be ameliorated by reducing the emphasis; it is certainly a novel prognostic indicator that should be taken into consideration in clinical decision-making, together with the genetic and clinical features of the disease. It is most certainly not the only factor “guiding treatment decisions”, nor possibly the most important. This is clearly stated in the Introduction, less so in the "short summary" and "Discussion" parts.
- page 3, line 137: “All patients had de novo disease”. But later “three quarters of the patients were diagnosed with primary AML and one quarter “were” diagnosed with secondary AML”. Please note that “de novo” usually refers to AML arising without prior hematological disorders (opposite to “secondary AML”); did the Authors mean “untreated” ? (that is, was the analysis done at diagnosis on samples before treatment?).
Comments on the Quality of English LanguageOverall, the language is clear and I had no difficulty in understanding. Minor lessical errors are present (being a non-native English speaker myself, I sympathise), and might easily be corrected. E.g. page 3, line 137: "three quarters of the patients were diagnosed with primary AML and one quarter “were” diagnosed with secondary AML”
Reviewer 2 Report
Comments and Suggestions for Authors
1. What do you think if you take bone marrow cases is there any difference in the expression of CD276 as compared with PBMC.
2. What type of correlation you have found CD276 linked with CD33, CD34, CD38, CD117 in your data.
3. What is the mean blast percentage in the cases you have included in this manuscript?
4. Is CD276 expression also linked with neutrophils, basophils, eosinophils, ldh parameters as well.
Reviewer 3 Report
Comments and Suggestions for Authors
In this study, the authors used a novel B7-H3 antibody, targeting a membrane distal epitope (8H8) of the B7-H3 protein they previously developed, for flow cytometric analysis of B7-H3 expression in 77 acute myeloid leukemia (AML) patients. B7-H3 expression was detected in 62.3% of AML patients and was higher in the monocytic French-American-British (FAB) M5 group and in intermediate and poor risk patients. Receiver operating characteristics (ROC) results showed that high B7-H3 expression was associated with shorter overall survival (OS) and progression-free survival (PFS).
This study demonstrated the novel, proprietary 8H8 mAb can be used as a reliable tool for immunophenotyping of AML patient samples and identified B7-H3 as a prognostic marker for AML patients.
In overall, the study is well-designed and valuable for clinical diagnosis of AML. The manuscript is well presented and organized. Only one mistake needed to be corrected: In the line of 70, “expression to in order improve risk stratification of AML patients” should be corrected to “expression in order to improve risk stratification of AML patients”. And please defined FAB in Abstract.
Round 2
Reviewer 1 Report
Comments and Suggestions for Authors
The authors properly addressed previous concerns about the Manuscript, which has been ameliorated.